# The Consumption of Food-Based Iodine in the Immediate Pre-Pregnancy Period in Madrid Is Insufficient. San Carlos and Pregnancy Cohort Study

**DOI:** 10.3390/nu13124458

**Published:** 2021-12-14

**Authors:** Verónica Melero, Isabelle Runkle, Nuria Garcia de la Torre, Paz De Miguel, Johanna Valerio, Laura del Valle, Ana Barabash, Concepción Sanabria, Inmaculada Moraga, Cristina Familiar, Alejandra Durán, Maria Jose Torrejón, Jose Angel Diaz, Martin Cuesta, Jorge Grabiel Ruiz, Inés Jiménez, Mario Pazos, Miguel Angel Herraiz, Nuria Izquierdo, Noelia Pérez, Pilar Matia, Natalia Perez-Ferre, Clara Marcuello, Miguel Angel Rubio, Alfonso Luis Calle-Pascual

**Affiliations:** 1Endocrinology and Nutrition Department, Hospital Clínico Universitario San Carlos and Instituto de Investigación Sanitaria del Hospital Clínico San Carlos (IdISSC), 28040 Madrid, Spain; veronica.meleroalvarez10@gmail.com (V.M.); irunkledelavega@gmail.com (I.R.); nurialobo@hotmail.com (N.G.d.l.T.); pazdemiguel@telefonica.net (P.D.M.); valeriojohanna@gmail.com (J.V.); lauradel_valle@hotmail.com (L.d.V.); ana.barabash@gmail.com (A.B.); csanabria786@hotmail.com (C.S.); inmamgg@hotmail.com (I.M.); cristinafamiliarcasado@gmail.com (C.F.); aduranrh@hotmail.com (A.D.); joseangelitodiaz@telefonica.net (J.A.D.); cuestamartintutor@gmail.com (M.C.); gajo_saru@hotmail.com (J.G.R.); i.jimenez.varas@gmail.com (I.J.); mario_pazos_guerra@hotmail.com (M.P.); mariapilar.matia@salud.madrid.org (P.M.); nataliaferre79@gmail.com (N.P.-F.); clara994@hotmail.com (C.M.); marubioh@gmail.com (M.A.R.); 2Centro de Investigación Biomédica en Red de Diabetes y Enfermedades Metabólicas Asociadas (CIBERDEM), 28029 Madrid, Spain; 3Medicina II Department, Facultad de Medicina, Universidad Complutense de Madrid, 28040 Madrid, Spain; 4Clinical Laboratory Department, Hospital Clínico Universitario San Carlos and Instituto de Investigación Sanitaria del Hospital Clínico San Carlos (IdISSC), 28040 Madrid, Spain; mjosetorrejon@gmail.com; 5Gynecology and Obstetrics Department, Hospital Clínico Universitario San Carlos and Instituto de Investigación Sanitaria del Hospital Clínico San Carlos (IdISSC), 28040 Madrid, Spain; maherraizm@gmail.com (M.A.H.); nuriaizquierdo4@gmail.com (N.I.); nperezp@salud.madrid.org (N.P.)

**Keywords:** based-food iodine intake, materno-fetal outcomes, food patterns, dairy products, shellfish

## Abstract

A pre-gestational thyroid reserve of iodine is crucial to guarantee the increased demand for thyroid hormone production of early pregnancy. An iodine intake ≥150 µg/day is currently recommended. The objective of this study was to assess average pre-gestational food-based iodine consumption in pregnant women at their first prenatal visit (<12 gestational weeks), and its association with adverse materno-fetal events (history of miscarriages, early fetal losses, Gestational Diabetes, prematurity, caesarean sections, and new-borns large/small for gestational age). Between 2015–2017, 2523 normoglycemic women out of 3026 eligible had data in the modified Diabetes Nutrition and Complication Trial (DNCT) questionnaire permitting assessment of pre-gestational food-based iodine consumption, and were included in this study. Daily food-based iodine intake was 123 ± 48 µg, with 1922 (76.1%) not reaching 150 µg/day. Attaining this amount was associated with consuming 8 weekly servings of vegetables (3.84; 3.16–4.65), 1 of shellfish (8.72; 6.96–10.93) and/or 2 daily dairy products (6.43; 5.27–7.86). Women who reached a pre-gestational intake ≥150 µg had lower rates of hypothyroxinemia (104 (17.3%)/384 (21.4%); *p* = 0.026), a lower miscarriage rate, and a decrease in the composite of materno-fetal adverse events (0.81; 0.67–0.98). Reaching the recommended iodine pre-pregnancy intake with foods could benefit the progression of pregnancy.

## 1. Introduction

Thyroid hormone (TH) is essential for embryo/fetal development [1,2]. Up until the 16–18th week of gestation, the only source of TH in the embryo/fetus is the maternal thyroid [1,3,4]. To meet the needs of pregnancy, the maternal thyroid must increase free T4 (FT4) secretion, particularly in the 1st trimester of gestation. During this time, circulating FT4 maternal levels actually rise, with a maximum between 7–11 gestational weeks (GW), in spite of passing TH to the embryo, and increasing levels of thyroid hormone binding protein. An insufficient production of TH by the maternal thyroid can result in severe neurocognitive delay [1]. Maternal thyroid hypofunction, albeit mild, is also associated with both maternal and fetal complications in pregnancy and the peripartum period. These include preterm delivery, early stage fetal losses, miscarriages, perinatal death/stillbirths, postpartum hemorrhage, fetal distress during labor, preeclampsia and higher risk of caesarean section [4,5].

Iodine is essential for TH synthesis [4,6]. Adequate iodine stores as well as an increase in iodine intake are needed to permit the physiological increase in maternal thyroid FT4 secretion characteristic of early stages of pregnancy [1]. However, dietary iodine deficiency is a relevant problem worldwide [7,8,9]. In fact, iodine deficit is still the most important cause of preventable neurocognitive dysfunction and motor disability in offspring [8,10,11,12,13,14,15]. To prevent iodine deficiency, the WHO recommends a daily iodine intake of 150 µg from foods and from iodized salt in non-pregnant adults, increasing to 250 µg/day during gestation, the latter requiring the use of iodine supplements as recommended by the WHO in some regions [16]. The importance of pre-gestational iodine ingestion has been shown by a study in which the use of iodized salt for a minimum of 2 years before pregnancy, enabling adequate thyroid stores of iodine, permitted better Thyroid stimulating Hormone (TSH) and/or FT4 levels than those found not only in women presenting an insufficient iodine intake before and throughout pregnancy, but also in women starting iodine supplements during pregnancy, without having assured adequate pre-gestation iodine consumption [17].

In the general population, and in women of childbearing age, the main sources of iodine are foods rich in iodine and iodized salt [18,19]. Among the former are fish, shellfish and seaweeds, as well as dairy products and several vegetables [13,20,21,22,23,24,25,26]. Although the use of iodized salt is recommended by the WHO, not all countries have established on-going campaigns to encourage its systematic use. Spain is an example of a country where a campaign was carried out for only a few months, more than 20 years ago. Although the WHO classified Spain as a country with a mild iodine deficiency in 2004 [27], it is currently considered an iodine-sufficient country [28,29]. However, iodine consumption varies in different regions [28,30,31,32,33,34].

Yet according to the results of a recent population-based study [35], the median urinary iodine in Spanish women between 18 and 50 years of age ranges from 114–119 µg/day, reflecting an adequate iodine intake (≥150 µg), since 90% of ingested iodine is excreted in the urine [36] but it may not be enough for women of childbearing age, especially in a pre-pregnancy situation. Furthermore, more than 30% of the women studied had a urinary iodine level below 50 µg/day, and less than 45% consumed iodized salt. To compound the problem, the use of iodized salt does not guarantee an adequate supply of iodine either. A recent FACUA (Federación de Asociaciones de Consumidores y Usuarios de Andalucía) [37] report, an Andalusian consumer organization (data not available) analysed 20 different classes of iodized salt. Most had a lower iodine content than what was given on the label. Iodine intake can therefore not be assured even when iodized salt is regularly used, highlighting the importance of food-based iodine consumption.

The primary objective of this study is to evaluate the average pre-gestational food-based iodine consumption in pregnant women studied at the onset of gestation. The results would permit knowing the number of women reaching the recommended pre-gestational daily iodine intake of 150 of µg with foods alone. The first secondary objective is to determine the eating pattern(s) associated with achieving the aforementioned daily food-based iodine consumption. The second one is to assess whether there is an association between not attaining said food-based iodine intake and the development of adverse materno-fetal outcomes.

## 2. Material and Methods

### 2.1. Experimental Design and Ethic Statement

The Hospital Clínico San Carlos is a public hospital located in the central area of Madrid. The hospital provides healthcare for a reference population of 445,000 inhabitants. Prenatal screening consultations of all pregnant women in the reference population are located in the hospital, where the women undergo the first ultrasound and the screening test to detect the risk of chromosomal alterations between 9 and 12 gestational weeks (GW). The current study involves the participants of our hospital-based cohort of pregnant women screened for gestational diabetes mellitus (GDM), from January 2015 to December 2017 (the San Carlos Cohort for the prevention of GDM ISRCTN84389045, ISRCTN13389832 and ISRCTN16896947).

It was approved by the Clinical Trials Committee of the Hospital Clínico San Carlos (CI 13/296-E, CI16/442-E, and CI 16/316), and conducted according to Declaration of Helsinki). All women signed the informed consent at the onset of the trial.

### 2.2. Participants

Women participating in the study met the following inclusion criteria: a fasting blood glucose (FBG) < 92 mg/dL, an age ≥ 18 years. Exclusion criteria were: having a multiple gestation, a gestational age (GW) > GW14 and/or intolerance/allergy to nuts or extra virgin olive oil (EVOO), or any medical conditions or pharmacological therapy that could compromise the follow-up program.

A total of 3026 consecutive women, who attended the prenatal screening visit before GW12, with a singleton pregnancy and without prior fertility treatment or known diseases, were initially eligible for the study. Women with pre-gestational thyroid disease and/or pre-gestational levothyroxine treatment (*n* = 135) and women without valid data available from the semi-quantitative questionnaire of frequency of food consumption (*n* = 368) were excluded. A total of 2523 women were included for analysis. The clinical characteristics of the included women are shown in Table 1.

### 2.3. AΠssessment of Food-Based Iodine Consumption

The semi-quantitative questionnaire used to register the number of weekly servings from different food groups was an adaptation of the Diabetes Nutrition and Complication trial (DNCT) questionnaire [38]. The DNCT questionnaire consists of 15 items, 3 of which provide information on physical activity (the physical activity score), and 12 of which evaluate food-frequency consumption (the Nutrition score). The questionnaire permits 3 options, with Option A (value +1) the most favourable, Option B (value 0) expresses an intermediate value, and Option C (value −1) the least favourable. In addition to evaluating eating patterns categorically, the number of weekly servings from 20 different food groups and minutes/week of sports activity of at least moderate intensity were collected. The questionnaire includes vegetables, fruits, nuts, white fish, fatty fish, canned fish, and shellfish. However, the specific subtype of food was not registered. Hence, to more precisely estimate iodine consumption, the following approximation was carried out. A total of 200 women chosen consecutively were evaluated to calculate the average iodine intake from the servings of each food group. For this survey, the first 50 consecutive women attending the prenatal clinic consultation before GW12 at the beginning of each quarter of the calendar year were included to reflect the seasonal character of many fruits, vegetables, fish, and shellfish. Food subtypes were specified. Foods whose iodine content was low, and contributed less than 1% of total iodine intake, were not tabulated in the study. Once the average composition of a serving from each food group was estimated, the calculation of iodine content was made using data from the Spanish Consumer Office, and tables of composition of iodine in food as reference [39,40]. Data are provided in Appendix A. The use of iodine supplements and the intake of iodized salt were not included for the estimation of pre-gestational food-based iodine consumption.

Simultaneously, the 14-point Mediterranean Diet Adherence Screener (MEDAS) questionnaire [41] was applied. Both questionnaires were conducted by a trained dietician at the first gestational visit, between 9 and 12 GW. At this time, participants retrospectively described their pre-pregnancy diet and physical activity. From these 2 questionnaires, the MEDAS and Nutrition scores were obtained. Both the MEDAS and DNCT questionnaires had been previously used in studies on the prevention of gestational diabetes. 

### 2.4. Data Collection

At the first visit, sociodemographic information such as age, ethnicity, educational level, employment status, and parity was collected. A family history of metabolic disease (type 2 DM, and each component of the metabolic syndrome), smoking habit, and habitual consumption of iodized salt, as well as the use of supplements with iodine (with or without folic acid), were recorded as clinical data. In addition, daily food-based iodine consumption in the pre-pregnancy period was also registered at this first appointment. Anthropometric data, such as current body weight (taken with light clothes and without shoes) and height (taken without shoes), to obtain body mass index (BMI) were collected. Declared body weight was also registered. Blood pressure was measured with an adequate armlet (Omron 705IT, Omron Global, Kyoto, Japan) when the participants had been seated for 10 min.

### 2.5. Follow-Up during Gestation

To evaluate the relationship between food-based iodine intake and materno-fetal adverse events, the following parameters were evaluated: a prior history of miscarriages and fetal losses occurring before GW18, prematurity (new-borns before GW37), development of pre-eclampsia (blood pressure greater than 140 mmHg and albuminuria from GW20 on) and diagnosis of gestational diabetes mellitus (GDM) applying the IADPSG diagnostic criteria, the form of delivery as vaginal or caesarean section, the number of newborns small for gestational age (SGA) and large for gestational age (LGA) considered as <10 percentile and >90 percentile, respectively, were analysed in this study. The composite materno-fetal adverse outcomes (CMFAO) variable analysed the risk of suffering from any adverse event previously described (miscarriage history, early fetal loss, preeclampsia, prematurity, caesarean section, GDM and SGA/LGA new-borns) and its association with reaching a food-based intake of ≥150 µg of iodine/day in pre-pregnancy women.

A diagnosis of hypothyroidism and/or hypothyroxinemia was also registered. Gestational subclinical hypothyroidism was defined as a TSH level ≥ 2.5 µIU/mL, and hypothyroxinemia as a FT4 ≥ 7.5 pg/mL during the first trimester of pregnancy.

### 2.6. Biochemical Data

To evaluate the biochemical markers corresponding with the baseline characteristics of women, an early-morning blood sample was collected after an 8–10 h fast. For this study, the following parameters were determined: fasting serum glucose (glucose oxidase); fasting serum insulin; homeostasis assessment model for insulin resistance (HOMA); total cholesterol and triglyceride levels (enzymatic); thyroid function was assessed by determining circulating TSH and FT4 levels. 

Fasting serum glucose was measured with a method standardized by the International Federation of Clinical Chemistry and Laboratory Medicine, using ion-exchange high-performance liquid chromatography in gradient, with a Tosoh G8 analyser (Tosoh Co., Tokyo, Japan). Fasting serum insulin (FSI) was measured by a chemiluminescence immunoassay in an IMMULITE 2000 Xpi (Siemens, Healthcare Diagnostics, Munich, Germany). Homeostasis assessment model for insulin resistance (HOMA) was calculated as glucose (mmol/L) × insulin (µUI/mL)/22.7). Total cholesterol and triglyceride levels (enzymatic) were analysed. TSH was measured by a 3rd generation sandwich-chemiluminescence immunoassay with magnetic particles using human TSH mouse monoclonal antibodies in a DXI-800^®^ (Beckman–Coulter). The manufacturer’s stated normal range for non-pregnant adults is 0.38–5.33 μIU/mL, with a sensitivity of 0.01 μIU/mL, intraassay coefficient of variation (CV) < 10%, and range 0.01–50.0 μIU/mL. Intra-assay CVs are 4.9% for a concentration of 0.69 μIU/mL, 5.8% for 5.47 μIU/mL, and 6.2% for 29 μIU/mL. FT4 was measured by a 2-step competitive-chemiluminescence immunoassay with paramagnetic particles, in a DXI-800^®^ (Beckman Coulter). The manufacturer’s stated normal range for non-pregnant adults is 5.8–16.4 pg/mL, with a sensitivity of 2.5 pg/mL, range: 2.5–60 pg/mL. Intra-assay CVs are 7.8% for a concentration of 8.5 pg/mL, 5.7% for 22.9 pg/mL and 4.3% for 43 pg/mL. To determine total cholesterol and triglycerides, a colorimetric method with CHOD-PAP and GPO-PAP, respectively, was used.

An External Quality Guarantee Program of the SEQC (Sociedad Española de Química Clínica) evaluates the quality of the methods monthly.

## 3. Statistical Analysis

Data are expressed as Mean + SD, median (Q1–Q3) or as number and percentage. For the comparison of continuous variables, Fisher’s test was used for unpaired data, and the chi-square test was used for categorical variables. Pre-gestational food-based iodine consumption was analysed. The use of iodine supplements and/or the consumption of iodized salt were not included in the analysis. Results were divided into 2 groups above and below the daily recommendation for pre-gestational consumption (150 µg/day). To explore the relationship between each lifestyle habit and consumption of at least 150 µg/day of iodine, a generalized linear binary logistic regression model was performed. The dependent variable was used to analyse a minimum iodine consumption of 150 µg/day versus a lower one. The reference value was 0 when iodine intake was less than 150 µg, whereas an intake ≥150 µg/day was taken as value 1. Twenty items were selected, according to their iodine content, as predictors of reaching an iodine consumption of at least 150 µg/day. The consumption of sixteen foods were analysed: vegetables, nuts, EVOO, fatty fish, canned fish, white fish, seafood, whole grains, dairy products (skimmed, semi-skimmed, whole and fortified), processed meats, mayonnaise, and dark chocolate. One item was for sports activity (registered in minutes of moderate physical activity) and 3 items for physical activity score. Nutrition score (12 items) and MEDAS score (14 items) were analysed. For each of the selected items, servings per week and the minutes per week of sports activity, were graduated on an increasing scale with the scores of the questionnaires used. Missing values were excluded from the analysis. The associations between food-based iodine intake per day as categorical variables (>150 µg/day vs. <150 µg/day) and gestational adverse outcomes were studied by the chi-squared test. The magnitude of association between food-based iodine per day intake >150 µg/day vs. <150 µg/day, with the latter as reference group, and gestational outcomes were evaluated using the odds ratio (OR) and 95% confidence interval (95%CI). Analysis was adjusted for age, parity, and smoking status.

## 4. Results

The mean daily iodine intake in the analysed women was 123 ± 48 µg/day, and the median (Q1–Q3) was 119 (90–140) µg/day. Of the 2523 analysed women, 1922 (76.1%) did not reach the recommended 150 µg of daily iodine intake from foods alone. Women who did not reach 150 µg of food-based iodine per day were younger, were more often multiparous, and had a lower educational status as compared to those who did. The former were more often active smokers, either up to the moment they became aware of pregnancy, or prior to as well as during pregnancy. The percentage of women with hypothyroxinemia was higher in those who did not reach the food-based iodine cut-off point. The consumption of iodized salt was also lower in the group of women who did not reach the daily 150 µg of food-based iodine. No significant differences were observed between both groups when comparing the other variables studied. These data are shown in Table 2.

The number of the predictive cut-off points of weekly servings of each food analysed, the scores of the MEDAS and DNCT questionnaires in association with a daily iodine intake of 150 µg are shown in Table 3.

After applying the binary logistic regression model, the probability of meeting the recommended iodine intake of 150 µg/day (OR; 95% CI) was associated with the consumption of at least 1 weekly serving of shellfish (8.72; 6.96–10.93), 2 daily servings of dairy products (6.43; 5.27–7.86), 8 weekly servings of vegetables (3.84; 3.16–4.65), 3 weekly servings of canned fish (3.07; 2.50–3.77), and/or 7 weekly servings of semi-skimmed dairy (3.02; 2.51–3.65). The MEDAS and Nutrition Score also yielded significant results when they were associated with the consumption of at least 150 µg of daily iodine intake. The physical activity score and the practise of at least 150 min weekly of sports-related activities was also statistically significant (2.45; 1.38–4.35). The results are displayed in Figure 1.

There was a tendency towards a lower rate of maternal-fetal adverse events among women who reached the recommended daily food-based iodine intake of 150 µg. There was a significantly lower rate of early fetal loss before GW18 in women who reached the aforementioned daily iodine intake when compared to those who did not ((3.1%) vs. (4.0%) respectively; *p* = 0.042). Furthermore, the rate of the composite variable of maternal-fetal adverse events was significantly lower in women who achieved the recommended pre-gestational iodine intake based on foods versus those who did not (205 (34.1%) vs. 747 (38.9%) respectively; *p* = 0.018). These data are shown in Table 4.

Women with a pre-pregnancy daily food-based iodine consumption of at least 150 µg showed a decrease in the rates of fetal losses < 18 GW (OR; 0.77; 0.39–1.03) as compared with those with a lower iodine food-based intake. In addition, the former women presented a 19% reduction in the risk of suffering at least one adverse event (composite variable) as compared with the latter, with an OR of 0.81 (0.81; 0.67–0.98). The data are given in Figure 2.

## 5. Discussion

A high percentage of the women in the current study (76.1%) did not reach the recommended pre-gestational level of iodine intake (≥150 µg) with foods alone. Additionally, only 38.6% of these women would have increased their iodine consumption through the pre-gestational use of iodized salt. Furthermore, less than 9% of women initiated iodine supplements before gestation. A recent survey conducted in the Norwegian population also detected a low food-based iodine intake in close to 50% of the women of fertile age studied [42]. Another Norwegian study assessing the food-based iodine consumption found that 74% of women had an average of food-based iodine intake lower than recommended during pregnancy [43], and less than 5% reached the daily iodine consumption recommended by the WHO in pregnancy [16] in the absence of iodine supplementation [44]. However, to our knowledge, the current study is the first to evaluate the average food-based iodine intake in the immediate pre-gestational period. Furthermore, pre-gestational iodine consumption corresponded with early pregnancy iodine intake in the vast majority of women studied, with iodine supplementation starting only upon diagnosis of pregnancy based on food and iodized salt consumption; almost half of the women we have studied would not have attained the recommended level of pre-gestational iodine intake. 

Approximately 25% of the women analysed in the current study did reach adequate levels of pre-gestational iodine consumption through dietary foods. In these women, at least 8 weekly servings of vegetables, 3 servings of canned fish, 1 serving of shellfish and/or 2 daily servings of dairy products contributed the most towards attaining a daily iodine intake ≥ 150 µg. The importance of assuring food-based iodine consumption is highlighted by the variability in the iodine content of that has been detected in available iodized salt products [37]. In fact, both packaging and weather conditions can substantially reduce the iodine content of commercially available iodized salt [45]. In addition, estimation of an individual’s salt consumption can be difficult, since the exact amount of salt an individual uses is variable, and quantification challenging. However, our study indicates that requisite pre-gestational iodine intake can be assured with foods alone, without depending on the use of “theoretically” iodized salt. Programs recommending universal consumption of iodized salt are extremely important, particularly if quality controls of the marketed on-the-shelf product are in place [46].

A higher percentage of the analysed women with an adequate food-based iodine intake regularly performed moderate physical activity. The group with a higher food-based iodine consumption could therefore have a healthier lifestyle. In fact, the women who performed physical exercise on a regular basis included less smokers, had a healthier diet, and a greater food-based iodine consumption than those who did not regularly exercise. 

The women who assured the requisite pre-gestational iodine intake with foods presented a lower rate of spontaneous miscarriage, and a lower rate of the composite variable of adverse materno-fetal events than those who did not. Iodine deficiency during pregnancy has been found to be related with materno-fetal complications, such as higher rates of Neonatal Intensive Care Unit (NICU) admissions and preterm delivery [47]. Moderate iodine deficiency is associated with a higher risk of miscarriage, preterm labor and low birth weight. Similar studies, using food-based iodine as reference, also detected that iodine deficiency is associated with a decrease in fetal growth, a higher risk of suffering preterm delivery and low fecundity and preeclampsia [44]. Even studies in mildly iodine-deficient regions have found that an adequate iodine status during pregnancy can decrease the rates of perinatal/infant mortality [48]. However, to our knowledge, the current study is the first to detect an association between an adequate food-based iodine consumption in the pre-pregnancy period and a reduction in the rate of the materno-fetal adverse events. 

Assuring a sufficient food-based iodine intake could be beneficial for pregnancy by permitting an adequate maternal thyroid response to pregnancy. Women with iodine insufficiency, albeit mild, do not present the physiological increase in circulating FT4 levels seen early in pregnancy [1]. In fact, in conditions of iodine insufficiency, FT4 levels can be low, without a corresponding rise in TSH levels, although the latter can also occur [1]. Furthermore, several authors have found isolated hypothyroxinemia, when occurring early in pregnancy, to be associated with adverse maternal/fetal outcomes [4,17,49,50,51,52,53,54,55]. In the current study, we detected a higher rate of low FT4 levels in the women who did not attain a pregestational (and therefore early gestational) food-based iodine intake of 150 µg or higher as compared with those who did (384 (21.4%) versus 104 (17.3%), *p* = 0.026, respectively). Hypothyroxinemia is likely an important contributor to the negative effect of a lower food-based iodine intake on the evolution of gestation [56].

Our study has limitations. Although urinary iodine is the most accurate way to determine iodine consumption, considering that the study was carried out at the first gestational visit, when the women studied were already using iodine supplements, this data would not reflect pre-pregnancy iodine consumption. Secondly, food frequency questionnaires can be inaccurate, particularly when retrospective. However, the questionnaire used was previously validated, and applied by a trained professionals. Lastly, the calculation of the iodine content was estimated based on the average iodine content of foods, number and size of servings, all of which are subject to variability and only 200 women, 50 per quarter, were studied for a detailed analysis of the specific food subtypes consumed. However, region-specific micronutrient food content tables were used. The intake of iodized salt was not included for the estimation of pre-gestational food-based iodine consumption. The iodine content of iodized salt is highly variable and is not reflected in the labelling. The salt consumption in general population is also extremely variable, taking into account also that >78% women work outside the home and do not make meals in their own home. This would make it very difficult to quantify accurately the consumption of iodine from iodized salt. Our study only considers the food-based iodine content, which more accurately reflects the importance of long-term iodine consumption prior to pregnancy. Although the rate of insufficient pregestational iodine intake could be reduced, this rate would still be important.

## 6. Conclusions

The current study finds that the pre-gestational food-based iodine consumption in our population is insufficient to assure an adequate iodine status in the absence of additional iodine sources. Fewer than 25% of the pregnant women evaluated had attained the recommended intake of ≥150 µg daily before pregnancy with foods alone. The women that achieved this level of food-based iodine consumption showed a reduced risk for spontaneous miscarriage, as well as for a composite of adverse materno-fetal events when compared with those eating foods with a lower iodine content. We believe that an optimal consumption of iodine-containing foods in women of childbearing age could help guarantee an adequate maternal iodine reserve, favoring a normal progression of pregnancy. Therefore, an iodine-sufficient diet should be encouraged in these women. Further studies are needed to assess whether this type of diet confers additional benefits to offspring.

## Figures and Tables

**Figure 1 nutrients-13-04458-f001:**
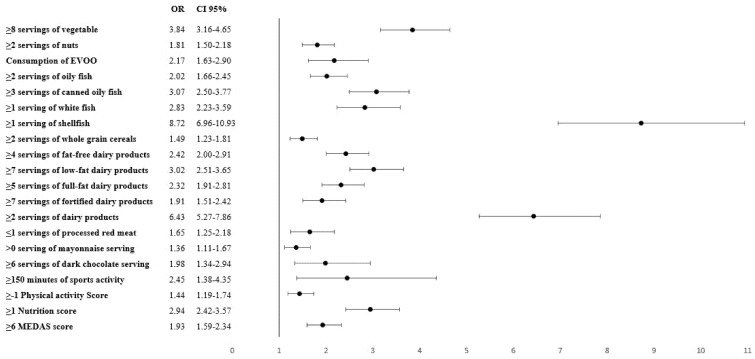
Probability of reaching a pre-gestational food-based iodine intake ≥ 150 µg/day and its association with specific food and servings. OR: Odds ratio; CI: confidence interval; EVOO: extra virgin olive oil; MEDAS: Mediterranean Diet Adherence Screener. All servings are weekly except for the consumption of EVOO and dairy products (without specifying fat content) which are daily.

**Figure 2 nutrients-13-04458-f002:**
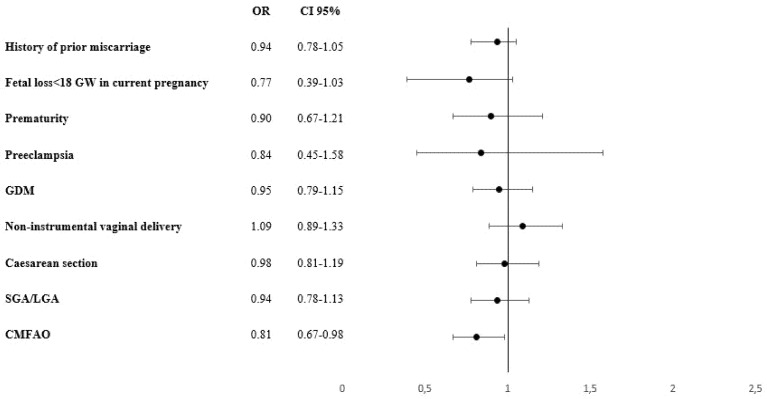
Risk for suffering at least one maternal-fetal adverse event and its association with the recommended daily iodine intake of 150 µg. GDM: Gestational Diabetes Mellitus. CMFAO: Composite Materno-foetal Adverse outcomes: at least one event of miscarriage history, early foetal loss, preeclampsia, prematurity, caesarean section, GDM and SGA: small for gestational age/LGA: large for gestational age newborns. OR: Odds ratio; CI: confidence interval.

**Table 1 nutrients-13-04458-t001:** Baseline characteristics of the women studied.

N	2523
Age	32.64 ± 5.19
Family history of: Type 2 diabetes/MetS(>2 components)	113 (4.5%)/560 (22.2%)
History of Prior Miscarriage	838 (33.2%)
Ethnicity: Caucasian/Hispanic	1530 (60.6%)/789 (31.3%)
Primiparous/multiparous	1081 (42.8%)/1432 (56.7%)
University or Technical degree	1646 (65.3%)
Works outside of the home	1969 (78%)
Smoker: until pregnancy/currently	320 (12.7%)/208 (8.2%)
BW (kg)	63.8 ± 11.4
BMI (kg/m^2^)	24.14 ± 4.09
SBP (mmHg)	108.83 ± 10.34
DBP (mmHg)	67.17 ± 8.79
FSG (mg/dL)	80.31 ± 6.07
FSI (mlU/L)	18.29 ± 21.15
HOMA-IR	3.65 ± 4.28
Cholesterol (mg/dL)	173.02 ± 30.96
Triglycerides (mg/dL)	81.48 ± 37.83
TSH (µIU/mL)	2.04 ± 1.53
FT4 (pg/mL)	8.64 ± 1.49
Pre-pregnancy Nutrition Score	0.32 ± 3.12
Pre-pregnancy MEDAS score	4.89 ± 1.74
Pre-pregnancy Physical activity score	−1.83 ± 0.96
Pre-pregnancy supplements:	
Iodine supplement	22 (0.9%)
Folic acid + iodine supplement	198 (7.8%)
Pregnancy supplements:	
Iodine supplement	127 (5.0%)
Folic acid + iodine supplement	1313 (52.0%)
Use of iodized salt	1021 (40.5%)

Data are Mean ± SD or number (%). METS: Metabolic Syndrome. BW: body weight; BMI: body mass index; SBP: systolic blood pressure; DBP: diastolic blood pressure; FSG: fasting serum glucose; FSI: fasting serum insulin; HOMA-IR: Homeostatic model assessment; MEDAS Score: 14-point Mediterranean Diet Adherence Screener; TSH: thyroid stimulating hormone; FT4: free thyroxin.

**Table 2 nutrients-13-04458-t002:** Characteristics of the women studied according to pre-gestational food-based iodine consumption.

	Iodine Intake < 150 µg/Day	Iodine Intake ≥ 150 µg/Day	*p*
N	1922 (76.1%)	602 (23.9%)	
Age	32.4 ± 5.3	33.4 ± 4.8	0.000
Miscarriage/GDM History	670 (34.9%)/61 (3.2%)	202 (33.5%)/24 (4.0%)	0.906
Caucasian Ethnicity	1166 (60.7%)	364 (60.6%)	0.166
Primipary	820 (42.8%)	261 (43.5%)	0.044
University degree	1214 (63.3%)	432 (71.8%)	0.003
Salaried work	1496 (77.9%)	474 (78.7%)	0.507
Family history of MetS	382 (19.9%)	121 (21.1%)	0.792
Smoker until/during pregnancy	255 (13.3%)/171 (8.9%)	65 (10.8%)/37 (6.1%)	0.010
Body Weight (Kg)	61.9 ± 11.5	61.5 ± 10.4	0.465
BMI (Kg/m^2^)	23.4 ± 4.1	23.1 ± 3.8	0.073
sBP (mm Hg)	108.9 ± 10.4	108.9 ± 10.1	0.915
dBP (mm Hg)	67.1 ± 8.9	67.3 ± 8.5	0.710
FSG (mg/dl)	80 ± 6	80 ± 6	0.435
FPI (µIU/mL)	18 ± 21	18 ± 21	0.124
HOMA-IR	3.64 ± 4.30	3.69 ± 4.25	0.777
Total Cholesterol (mg/dl)	172 ± 30	175 ± 33	0.914
Triglycerides (mg/dl)	81 ± 34	84 ± 46	0.092
TSH µIU/mLTSH > 2.5 µIU/mL	2.13 ± 1.41 530 (27.6%)	2.02 ± 1.41 162 (27.0%)	0.1660.411
FT4 pg/mLFT4 < 7.5 pg/mL	8.56 ± 1.57 384 (21.4%)	8.67 ± 1.47 104 (17.3%)	0.1700.026
Use of Iodized salt	742 (38.6%)	280 (46.5%)	0.006

GDM: gestational diabetes mellitus; MetS: metabolic syndrome; BMI: body mass index; sBP: systolic blood pressure; dBP: diastolic blood pressure; FSG: fasting serum glucose; FPI: fasting plasma insulin; HOMA-IR: Homeostatic model assessment; TSH: thyroid stimulating hormone; FT4: free thyroxine. The consumption of iodized salt was not included in the analysis of pre-gestational food-based iodine consumption.

**Table 3 nutrients-13-04458-t003:** Rate of pre-gestational individual foods consumption and exercise per group of pre-gestational food-based iodine intake.

	Iodine Intake < 150 µg/Day	Iodine Intake ≥ 150 µg/Day	*p*
≥8 servings of vegetable	406/1922 (21.1%)	305/602 (50.7%)	0.000
≥2 servings of nuts	580/1922 (30.2%)	264/602 (43.9%)	0.000
Consumption of EVOO	1544/1922 (80.3%)	541/602 (89.9%)	0.000
≥2 servings of oily fish	470/1922 (24.5%)	238/602 (39.5%)	0.000
≥3 servings of canned oily fish	302/1922 (15.7%)	219/602 (36.4%)	0.000
≥1 serving of white fish	1262/1922 (65.7%)	508/602 (84.4%)	0.000
≥1 serving of shellfish	168/1922 (8.7%)	274/602 (45.5%)	0.000
≥2 servings of whole grain cereals	527/1921 (27.4%)	217/602 (36.0%)	0.000
≥4 servings of fat-free dairy products	576/1922 (30.0%)	306/602 (50.8%)	0.000
≥7 servings of low-fat dairy products	569/1922 (29.6%)	337/602 (56.0%)	0.000
≥5 servings of full-fat dairy products	900/1922 (46.8%)	404/602 (67.1%)	0.000
≥7 servings of fortified dairy products	248/1921 (12.9%)	133/602 (22.1%)	0.000
≥2 servings of dairy products	476/1921 (24.8%)	409/602 (67.9%)	0.000
≤1 servings of processed red meat	1583/1922 (82.4%)	532/601 (88.2%)	0.000
>0 serving of mayonnaise serving	1265/1921 (65.9%)	436/602 (72.4%)	0.001
≥6 servings of dark chocolate serving	70/1922 (3.6%)	42/602 (7.0%)	0.001
≥150 min of sports activity	28/1921 (1.5%)	21/600 (3.5%)	0.001
≥−1 Physical activity Score	583/1919 (30.4%)	232/601 (38.6%)	0.000
≥1 Nutrition score	791/1914 (41.3%)	404/599 (67.4%)	0.000
≥6 MEDAS score	574/1832 (31.3%)	267/570 (46.8%)	0.000

EVOO: extra virgin olive oil; MEDAS: 14-points Mediterranean Diet Adherence Screener; All servings are weekly except for the consumption of EVOO and dairy products (those without specifying fat content), which are daily.

**Table 4 nutrients-13-04458-t004:** Maternal-fetal adverse outcomes by pre-gestational food-based iodine intake.

	Iodine Intake < 150 µg/Day	Iodine Intake ≥ 150 µg/Day	*p*
History of prior miscarriage	672/1922 (35.1%)	201/602 (33.4%)	0.296
Fetal loss < 18 GW in current pregnancy	68/1922 (4.0%)	18/602 (3.1%)	0.042
Prematurity	104/1854 (5.6%)	28/584 (4.9%)	0.077
Preeclampsia	28/1854 (1.5%)	7/584 (1.2%)	0.374
GDM	367/1854 (19.8%)	109/584 (18.7%)	0.326
Non-instrumental vaginal delivery	1118/1854 (60.3%)	364/584 (62.3%)	0.230
Caesarean section	402/1854 (21.7%)	124/584 (21.2%)	0.764
SGA/LGA	401/1854 (21.6%)	104/584 (17.8%)	0.280
CMFAO	747/1922 (38.9%)	205/602 (34.1%)	0.018

GDM: gestational diabetes mellitus; SGA: small for gestational age; LGA: large for gestational age; CMFAO: Composite materno-foetal Adverse outcomes. The consumption of iodized salt was not included in the analysis of pre-gestational food-based iodine consumption.

## Data Availability

The data presented in this study are available in supplementary metadata. The data presented in this study are available on request from the corresponding author.

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
