# Peer review of "The Consumption of Food-Based Iodine in the Immediate Pre-Pregnancy Period in Madrid Is Insufficient. San Carlos and Pregnancy Cohort Study"

_nutrients, 2021, doi:10.3390/nu13124458_

Round 1

Reviewer 1 Report

Dear Authors, 

Thank you for this paper which reviews food-based iodine intake with the development of adverse materno-fetal outcomes. Please find below comments for your consideration:

Participants: please consider including a flow chart to show the how the 3026 eligible women reduced to the 2523 included in the study, demonstrating at which points and which data points were not complete and therefore excluded from the study. Is the study powered to detect change and associations between iodine intake and materno-fetal outcomes?

Table 1: it is interesting that 'smoker' was a question, but alcohol consumption was not included in the survey? Was there a reason for this? Could alcohol intake be a factor which should have been considered in this research - ie. healthier diets will include less alcohol intake?

Table 1: smoking rates and BMI are both high, is this information reflective of the general population?

Line 139: A total of 200 women chosen, could you please provide additional information on how these women were consecutively selected?  What impact does this have on the research? Limitations of the study?

Line 361: Study limitations, there are several limitations of this study which have not been outlined, including  -  the lack of urinary iodine samples to validate the FFQs. Gold standard for iodine (and sodium) intake is urinary intake analysis. Why were urine samples not collected? And if collected, why have they not been presented within this paper?

Author Response

Thank you very much for your kind comments and constructive suggestions. We agree with you on most points. We have made the corresponding changes to improve the article.

The changes applied are as follows:

  1. Participants: please consider including a flow chart to show the how the 3026 eligible women reduced to the 2523 included in the study, demonstrating at which points and which data points were not complete and therefore excluded from the study. Is the study powered to detect change and associations between iodine intake and materno-fetal outcomes?

Answer: We included in the text the reasons for exclusion. Since the 2 reasons for exclusion are straightforward, we do not think there is enough content to justify a flow chart.  However, we have clarified the text: “A total of 3026 consecutive women, who attended the prenatal screening visit before GW12, with a singleton pregnancy and without prior fertility treatment or known diseases, were initially eligible for the study. Women with pre-gestational thyroid disease and/or pre-gestational levothyroxine treatment (n=135) and women without valid data available from the semi-quantitative questionnaire of frequency of food consumption (n=368) were excluded. A total of 2523 women were included for analysis. The clinical characteristics of the included women are shown in Table 1”

As explained in the text lines 103-106, The current study involves the participants of our hospital-based cohort of pregnant women screened for gestational diabetes mellitus (GDM), from January 2015 to December 2017 (the San Carlos Cohort for the prevention of GDM ISRCTN84389045, ISRCTN13389832 and ISRCTN16896947).The power of the study to detect changes in the maternal-fetal composite variable and iodine consumption as a continuous variable was not estimated. Iodine consumption from food was estimated as a categorical variable, not a continuous one.

  1. Table 1: it is interesting that 'smoker' was a question, but alcohol consumption was not included in the survey? Was there a reason for this? Could alcohol intake be a factor which should have been considered in this research - ie. healthier diets will include less alcohol intake?

Answer: The DNCT questionnaire includes the categorical estimation of alcohol consumption. Pre-pregnancy data for alcohol consumption are as follows:

Tabla de contingencia

1 <150 VS 2 >150

Total

1,00

2,00

AlcoholpreG

>6/dia

Recuento

1

2

3

% dentro de 1 <150 VS 2 >150

,1%

,3%

,1%

0 o  4-6/dia

Recuento

1864

582

2446

% dentro de 1 <150 VS 2 >150

97,0%

96,7%

96,9%

1-4/dia

Recuento

56

18

74

% dentro de 1 <150 VS 2 >150

2,9%

3,0%

2,9%

Total

Recuento

1921

602

2523

% dentro de 1 <150 VS 2 >150

100,0%

100,0%

100,0%

Pruebas de chi-cuadrado

Valor

gl

Sig. asintótica (bilateral)

Chi-cuadrado de Pearson

3,041a

2

,219

Razón de verosimilitudes

2,473

2

,290

Asociación lineal por lineal

,065

1

,799

N de casos válidos

2523

a. 2 casillas (33,3%) tienen una frecuencia esperada inferior a 5. La frecuencia mínima esperada es ,72.

As can be seen, there are no differences between groups, and it does not influence our results. Therefore we do not refer to it in the paper

  1. Table 1: smoking rates and BMI are both high, is this information reflective of the general population?

Answer: These data reflect those of the general population, as regards smoking. During pregnancy, slightly less than 10% of the women were active smokers, whereas about 20% of the general adult population are smokers. The BMI is slightly higher than that referred for the general population.

  1. Line 139: A total of 200 women chosen, could you please provide additional information on how these women were consecutively selected?  What impact does this have on the research? Limitations of the study?

Answer: We have included the following sentence in the text (line 140):  “The first 50 consecutive women attending the prenatal clinic consultation before GW12 at the beginning of each quarter of the calendar year were included”.

Answer: This is also reflected in the limitations of the study, lines 363-365: “Food frequency questionnaires can be inaccurate, particularly when retrospective. However, the questionnaire used was previously validated, and applied by a trained professionals. Secondly, the calculation of the iodine content was estimated on the average iodine content of foods, number and size of servings, all of which are subject to variability”. We have added to limitations: “Only 200 women, 50 per quarter, were studied for a detailed analysis of the specific food subtypes consumed”.

  1. Line 361: Study limitations, there are several limitations of this study which have not been outlined, including  -  the lack of urinary iodine samples to validate the FFQs. Gold standard for iodine (and sodium) intake is urinary intake analysis. Why were urine samples not collected? And if collected, why have they not been presented within this paper?

Answer: The urinary iodine level was not determined. The primary objective of the study was the prevention of GDM as we have previously reflected in the inclusion criteria and the assessment of urinary iodine was not included. We have included a comment on the limitations (line 361). The determination of urinary iodine is a more sensitive way to estimate the iodine consumption of the 2-3 days prior to obtaining the sample, but considering that the iodine consumption is estimated in the first prenatal visit, when women were already taking iodine supplements, this parameter would not reflect the pre-pregnancy consumption that is the objective of this study.

Line 361: “Although urinary iodine is the most accurate way to determine iodine consumption, considering that the study was carried out in the first gestational visit, when the attending women were already receiving iodine supplements, urinary iodine would not reflect pre-pregnancy iodine consumption”

Reviewer 2 Report

This is an assessment of predictors of preconception dietary iodine intake and associations of preconception dietary iodine intake in with gestational hypothyroxinemia and adverse pregnancy outcomes in a cohort of 2,523 Spanish women.

  1. Line 42 and throughout “embrio” should be “embryo.”
  1. Line 57: an updated reference which could be cited with regard to the epidemiology of iodine deficiency is: Zimmermann MB, Andersson M. GLOBAL ENDOCRINOLOGY: Global perspectives in endocrinology: coverage of iodized salt programs and iodine status in 2020. Eur J Endocrinol. 2021 Jun 10;185(1):R13-R21. doi: 10.1530/EJE-21-0171. PMID: 33989173
  1. Line 61, “the latter requiring the use of iodine supplements”: the WHO recommendations, which are cited here, currently only recommend iodine supplementation in pregnancy in some regions; this is not a universal recommendation.
  1. Line 75: with regard to the current iodine status of Spain, the WHO/IGN current scorecard could also be cited: https://www.ign.org/cm_data/IGN_Global_Scorecard_2021_7_May_2021.pdf
  1. Lines 77-78, “the median urinary iodine in Spanish women between 128 and 50 years of age ranges from 114-119 mcg/L, reflecting an inadequate iodine intake”: these median values are consistent with iodine sufficiency in populations of non-pregnant adults.
  1. Lines 136-137: why is dairy (one of the major contributors to iodine in the diet) not mentioned here?
  1. Line 180: the need for caesarean section is a fairly soft endpoint. How do results change if this is removed from the composite outcome?
  1. Lines 184-185, “gestational subclinical hypothyroidism was defined as a TSH level > 2.5 mIU/L”: This is an outdated definition and likely overestimated hypothyroidism in this sample. The currently accepted upper limit for TSH in gestation is 4.0 mIU/L (see the ATA guideline, reference #51). How was the lower limit for free T4 defined?
  1. Statistical Analysis, lines 216-237: how were analyses of associations between iodine intake and gestational outcomes assessed? What potential confounders were included in these analyses?
  1. Table 4: shouldn’t the proportions of cesarean sections and vaginal deliveries total 100% for each group?
  1. Lines 321-328: this discussion likely understates the effectiveness of universal salt iodization strategies. E.g., see the following: Dold S, Zimmermann MB, Jukic T, Kusic Z, Jia Q, Sang Z, Quirino A, San Luis TOL, Fingerhut R, Kupka R, Timmer A, Garrett GS, Andersson M. Universal Salt Iodization Provides Sufficient Dietary Iodine to Achieve Adequate Iodine Nutrition during the First 1000 Days: A Cross-Sectional Multicenter Study. J Nutr. 2018 Apr 1;148(4):587-598. doi: 10.1093/jn/nxy015. PMID: 29659964.
  1. Lines 361-366: a lack of ascertainment of urinary iodine concentrations could be included among the study limitations.
  1. For women who ingested iodine-containing supplements, what was the average daily iodine dose?

Author Response

Thank you very much for your kind comments and constructive suggestions. We agree with you on most points. We have made the corresponding changes to improve the article.

The changes applied are as follows:

  1. Line 42 and throughout “embrio” should be “embryo.”

Answer: Thank you for your suggestion. We have already corrected it.

  1. Line 57: an updated reference which could be cited with regard to the epidemiology of iodine deficiency is: Zimmermann MB, Andersson M. GLOBAL ENDOCRINOLOGY: Global perspectives in endocrinology: coverage of iodized salt programs and iodine status in 2020. Eur J Endocrinol. 2021 Jun 10;185(1):R13-R21. doi: 10.1530/EJE-21-0171. PMID: 33989173

Answer: Thank you for your suggestion. We have included this reference: “Zimmermann MB, Andersson M. GLOBAL ENDOCRINOLOGY: Global perspectives in endocrinology: coverage of iodized salt programs and iodine status in 2020. Eur J Endocrinol. 2021, 185, R13-R21. doi: 10.1530/EJE-21-0171”.

  1. Line 61, “the latter requiring the use of iodine supplements”: the WHO recommendations, which are cited here, currently only recommend iodine supplementation in pregnancy in some regions; this is not a universal recommendation.

Answer: We add in the phrase "in some regions". In Spain it is recommended by the Spanish Society of Gynecology and Obstetrics (SEGO). Line 61: “as recommended by the WHO in some regions”.

Line 75: with regard to the current iodine status of Spain, the WHO/IGN current scorecard could also be cited: https://www.ign.org/cm_data/IGN_Global_Scorecard_2021_7_May_2021.pdf

Answer: Thank you for your comments. We have included this reference: https://www.ign.org/cm_data/IGN_Global_Scorecard_2021_7_May_2021.pdf

  1. Lines 77-78, “the median urinary iodine in Spanish women between 128 and 50 years of age ranges from 114-119 mcg/L, reflecting an inadequate iodine intake”: these median values are consistent with iodine sufficiency in populations of non-pregnant adults.

Answer: We agree with your comments. We have modified the sentence to clarify concepts: “Yet according to the results of a recent population-based study [33], the median urinary iodine in Spanish women between 18 and 50 years of age ranges from 114-119 µg/day, reflecting an adequate iodine intake for non-pregnant adults since 90% of ingested iodine is excreted in the urine [34], but it may not be enough for women of childbearing age, especially in a pre-pregnancy situation. Furthermore, more than 30% of the women studied had a urinary iodine level below 50 µg/day.

  1. Lines 136-137: why is dairy (one of the major contributors to iodine in the diet) not mentioned here?

Answer: It is not mentioned because it is obtained directly from the DNCT questionnaire. Only the food groups that do not specifically include the specific type of food are referred to, which is why the sub-study is carried out to estimate the iodine content of a serving of these foods. As can be seen in the results, the consumption of daily products exerts an important effect.

  1. Line 180: the need for caesarean section is a fairly soft endpoint. How do results change if this is removed from the composite outcome?

Answer: We agree with your comment. However, cesarean section is seen as an adverse event, since it usually reflects, in our hospital,a loss of fetal well-being, or lack of progression of dilation. However, the rate of C.-sections does not affect the results. As can be seen in the figure, the rates are very similar in both groups: 21.7% vs 21.2%. It does not affect the results in the composite variable.

  1. Lines 184-185, “gestational subclinical hypothyroidism was defined as a TSH level > 2.5 mIU/L”: This is an outdated definition and likely overestimated hypothyroidism in this sample. The currently accepted upper limit for TSH in gestation is 4.0 mIU/L (see the ATA guideline, reference 57). How was the lower limit for free T4 defined?

Answer: When this study was carried out it was 2015-17, and at that time, based on the existing evidence, ATA 2011 and ETA 2014 guidelines recommended the use of this cut-off point.  Gestational subclinical hypothyroidism was defined as a TSH level > 2.5 mIU/L The new guides after our study, ATA 2017 are discussed in the discussion lines 355-360, hypothyroxinemia (FT4l< 7.5 pg/mL) corresponds with the 10th centile in the non-pregnant population in our laboratory, which is the cut-off point used in our protocol (57).

  1. Statistical Analysis, lines 216-237: how were analyses of associations between iodine intake and gestational outcomes assessed? What potential confounders were included in these analyses?

Answer: We have included the following in the text (line 237): “The associations between food-based iodine intake per day as categorical variables (>150 µg/day vs <150 µg/day) and gestational adverse outcomes were studied by the chi-squared test. The magnitude of association between food-based iodine per day intake >150 µg/day vs. <150 µg/day, with the latter as reference group, and gestational outcomes was evaluated using the odds ratio (OR) and 95% confidence interval (95%CI). Analysis was adjusted for age, parity and smoking status”.

  1. Table 4: shouldn’t the proportions of cesarean sections and vaginal deliveries total 100% for each group?

Answer: Vaginal deliveries are not instrumental. We have included this aspect in table 4 “Non-instrumental vaginal delivery”.

  1. Lines 321-328: this discussion likely understates the effectiveness of universal salt iodization strategies. E.g., see the following: Dold S, Zimmermann MB, Jukic T, Kusic Z, Jia Q, Sang Z, Quirino A, San Luis TOL, Fingerhut R, Kupka R, Timmer A, Garrett GS, Andersson M. Universal Salt Iodization Provides Sufficient Dietary Iodine to Achieve Adequate Iodine Nutrition during the First 1000 Days: A Cross-Sectional Multicenter Study. J Nutr. 2018 Apr 1;148(4):587-598. doi: 10.1093/jn/nxy015. PMID: 29659964.

Answer: Thank you for your comments. We have included this reference and sentence: (line 328). “Programs recommending universal consumption of iodized salt are extremely important, particularly if quality controls of the marketed on-the-shelf product are in place”. 44. Dold S, Zimmermann MB, Jukic T, Kusic Z, Jia Q, Sang Z, Quirino A, San Luis TOL, Fingerhut R, Kupka R, Timmer A, Garrett GS, Andersson M. Universal Salt Iodization Provides Sufficient Dietary Iodine to Achieve Adequate Iodine Nutrition during the First 1000 Days: A Cross-Sectional Multicenter Study. J Nutr. 2018, 148, 587-598. doi: 10.1093/jn/nxy015.

  1. Lines 361-366: a lack of ascertainment of urinary iodine concentrations could be included among the study limitations.

Answer: Thank you for your comments. It has been included in the limitations. Line 361: “Although urinary iodine is the most accurate way to determine iodine consumption, considering that the study was carried out at the first gestational visit, when the women studied were already using iodine supplements, this data would not reflect pre-pregnancy iodine consumption”.

  1. For women who ingested iodine-containing supplements, what was the average daily iodine dose?

Answer: The consumption of supplements with iodine only started within a few weeks of gestation. Daily iodine supplements for pregnancy in Spain contain 250 mcg. In any event, as reflected in Table 1, less than 9% of the women studied had initiated iodine supplementation before pregnancy. Furthermore, the daily intake of iodine from food is similar among women who take or who do not take supplements with iodine in each group. We had also reflected in the text the importance of long-term iodine consumption prior to pregnancy.

Lines 61-67: The importance of pre-gestational iodine ingestion has been shown by a study in which the use of iodized salt for a minimum of 2 years before pregnancy, enabling adequate thyroid stores of iodine, permitted better Thyroid stimulating Hormone (TSH) and/or FT4 levels than those found not only in women presenting an insufficient iodine intake before and throughout pregnancy, but also in women starting iodine supplements during pregnancy, without having assured adequate pre-gestation iodine consumption [17].

Round 2

Reviewer 2 Report

My prior comments have been thoughtfully addressed and I do not have additional concerns.

Author Response

Thank you very much for your kind comments. No questions asked, No answer required
